# Integration of whole genome sequencing and transcriptomics reveals a complex picture of the reestablishment of insecticide resistance in the major malaria vector *Anopheles coluzzii*

Victoria A. Ingham[1,2]*, Jacob A. Tennessen[3,4], Eric R. Lucas[1], Sara Elg[1], Henrietta Carrington Yates[1], Jessica Carson[1], Wamdaogo Moussa Guelbeogo[5], N'Fale Sagnon[5], Grant L. Hughes[1,6], Eva Heinz[1,7], Daniel E. Neafsey[3,4], Hilary Ranson[1]*

1 Vector Biology Department, Liverpool School of Tropical Medicine, Liverpool, United Kingdom,
2 Parasitology Unit, Universitätsklinikum Heidelberg, Heidelberg, Germany, 3 The Broad Institute, Cambridge, Massachusetts, United States of America, 4 Harvard TH Chan School of Public Health, Boston, Massachusetts, United States of America, 5 Centre National de Recherche et de Formation sur le Paludisme, Ougadougou, Burkina Faso, 6 Tropical Disease Biology, Liverpool School of Tropical Medicine, Liverpool, United Kingdom, 7 Department of Clinical Sciences, Liverpool School of Tropical Medicine, Liverpool, United Kingdom

* Victoria.ingham@uni-heidelberg.de (VAI); Hilary.ranson@lstmed.ac.uk (HR)

**Data Availability Statement:** Custom code written for this paper can be found at https://github.com/VictoriaIngham/Banfora_Paper. All sequence data

## Abstract

Insecticide resistance is a major threat to gains in malaria control, which have been stalling and potentially reversing since 2015. Studies into the causal mechanisms of insecticide resistance are painting an increasingly complicated picture, underlining the need to design and implement targeted studies on this phenotype. In this study, we compare three populations of the major malaria vector *An. coluzzii*: a susceptible and two resistant colonies with the same genetic background. The original colonised resistant population rapidly lost resistance over a 6-month period, a subset of this population was reselected with pyrethroids, and a third population of this colony that did not lose resistance was also available. The original resistant, susceptible and re-selected colonies were subject to RNAseq and whole genome sequencing, which identified a number of changes across the transcriptome and genome linked with resistance. Firstly, an increase in the expression of genes within the oxidative phosphorylation pathway were seen in both resistant populations compared to the susceptible control; this translated phenotypically through an increased respiratory rate, indicating that elevated metabolism is linked directly with resistance. Genome sequencing highlighted several blocks clearly associated with resistance, including the 2Rb inversion. Finally, changes in the microbiome profile were seen, indicating that the microbial composition may play a role in the resistance phenotype. Taken together, this study reveals a highly complicated phenotype in which multiple transcriptomic, genomic and microbiome changes combine to result in insecticide resistance.

has been deposited in SRA under accession PRJNA750256 and PRJNA764501.

**Funding:** This study was funded by an MRC Skills Development award to VAI [MR/R024839/1]. This project has been funded in whole or in part with Federal funds from the National Institute of Allergy and Infectious Diseases, National Institutes of Health, Department of Health and Human Services, under Grant Number U19AI110818 to the Broad Institute. EH and GLH were supported by the BBSRC (V011278/1). The funders had no role in study design, data collection and analysis, decision to publish, or preparation of the manuscript.

**Competing interests:** The authors have declared that no competing interests exist.

## Author summary

Insecticide resistance in major malaria vectors represents the single biggest threat to malaria control programs, which are heavily reliant upon insecticide-based interventions. Studying resistance using multi-omics approaches has proven difficult due to the use of susceptible comparator populations that have been colonised in a laboratory setting for decades, leading to substantial noise in the data due to differing genetic backgrounds. Here, we utilise a resistant *Anopheles coluzzii* population from Burkina Faso, a derived population that rapidly lost resistance over a 6-month period, and a population re-selected after loss of resistance to explore causative mechanisms of insecticide resistance. To determine the underlying cause of this phenotype, we use RNAseq, whole genome sequencing and lab-based validation to show changes in respiratory rate, wide-ranging genomic changes and alterations in the microbiome are linked to resistance in this population. These findings demonstrate the complexity of resistance and the challenges in utilising diagnostic markers for resistance in a field setting.

## Background

Over 80% of the reductions seen in malaria incidence since the turn of the century have been directly attributed to the use of insecticide-based vector control tools [1]. Despite these significant declines in malaria related morbidity and mortality, progress has stalled in the last three years, with evidence that malaria case numbers are again on the rise [2]. The plateau seen in malaria cases corresponds strongly with the spread of insecticide resistance in the major Anopheline vectors, allowing some mosquito populations to survive multiple exposures to key vector control interventions with no impact on their lifespan [3, 4]. Extremely high levels of insecticide resistance are now found in some settings due to the intense selection pressure caused by the use of relatively few public health insecticides and the use of the same classes of insecticides for controlling agricultural pests [2]. For example, insecticide treated bednets, the most widely utilised and most effective vector control tool, all contain insecticides from the pyrethroid class [2]. Furthermore, the use of pyrethroids in an agricultural setting adds additional selection pressures in the larval habitats, further reinforcing resistance to these chemistries [5]. To restore efficacy of bednets, next generation bednets containing pyrethroid insecticides and a different chemical class are now being distributed [6–8]. The second chemistries contained in the bednets either synergise the effects of the pyrethroid through targeting enzymes that break down the insecticide [6], act as a second slower acting insecticide [8] or sterilise female adult mosquitoes [7]. The efficacy of these different classes of nets will depend on the characteristics of the local vector population, with cross resistance [9, 10], or limited synergism [11] reported in some settings, that may compromise their efficacy in the field. For example, pre-treatment with the metabolic detoxification inhibitor PBO, now incorporated into bednets from multiple manufacturers, does not always result in full susceptibility [12]. Thus, an understanding of the causes of resistance is important for the development, evaluation, and implementation of new vector control tools.

Insecticide resistance in *Anopheles coluzzii*, one of the most important African malaria vectors, is multi-faceted and recent work has revealed previously unexplored mechanisms that are contributing to the pyrethroid resistance phenotype [13–15]. The first pyrethroid resistance mechanism to be described was target site resistance, these are single nucleotide polymorphisms found within the protein targeted by the insecticide rendering them less effective [16, 17]. Pyrethroid insecticides target the *para* gated sodium channel and several mutations in this

gene have been shown to contribute to increased resistance [16]. Another less well characterised resistance mechanism is the thickening of the cuticle [18]; this reduces penetrance of the insecticide and therefore likely results in lower cellular concentrations. Metabolic resistance to pyrethroids is largely conferred by specific cytochrome p450s, which have been shown to be highly up-regulated across multiple *Anopheles* populations [14] and actively metabolise pyrethroid insecticides [9, 19–21]. Recent work has demonstrated a key role for the chemosensory protein family in pyrethroid resistance [13], and other gene families with sequestration functions have also been implicated in this phenotype [14]. Taken together, the mosquito vector can make use of one or more of these mechanisms in parallel [22], with recent work demonstrating synergy of different mechanisms [23].

Insecticide resistance is further complicated by sub-lethal exposure to insecticides; this is especially important in areas of high resistance where mosquitoes may contact insecticide multiple times throughout their lifetime [3]. A number of targeted studies have shown induction of genes involved in metabolic resistance, potentially involving oxidative stress sensing pathways [24] and evidence points to induction of chemosensory proteins post-exposure [13]. Indeed, a recent induction study using sub-lethal pyrethroid exposure has shown large scale changes in the transcriptome including huge down-regulation of the oxidative phosphorylation pathway post-exposure [15]. Sub-lethal exposure is also important in the context of a changing microbiome, which has been shown to be modified upon exposure [25] and several bacterial species have been linked to the resistance phenotype [26, 27].

Transcriptomic studies on insecticide resistance are confounded by the use of a susceptible species that is lab adapted and has been kept in colony for decades as a comparator [14]. The differences in genetic backgrounds between the resistant population of interest and the susceptible control may identify differential expression attributed to the differing genetic backgrounds and not the resistance status of the mosquito. Further, whole genome sequence data in anopheline mosquitoes is limited, with the Ag1000G project representing the largest resource; however, very few sequenced samples have associated insecticide phenotyping data [28]. These factors make interpretation of big-data -omics in this field more difficult and each finding must be extensively validated in the lab, preventing fast identification of new or novel resistance mechanisms.

In this study we utilise a highly insecticide resistant population of *An. coluzzii* colonised in 2014 from Burkina Faso [22] which unexpectedly and rapidly lost resistance within a 6-month period. The susceptible colony reared from this population and the subsequent re-selection of the colony to full resistance in four generations allows a unique comparison of resistant and susceptible mosquitoes with the same genetic background. Here, we use RNAseq and single individual whole genome sequence data to identify changes within the mosquito's genome, transcriptome and microbiome contributing to the change in resistance phenotype. We show that pyrethroid resistance is associated with higher basal metabolism and numerous polymorphisms clustered on large haplotype blocks and we identify a number of divergent single nucleotide polymorphisms (SNPs) driving the phenotypic change. Finally, we show that changes in the microbiome composition are linked to the resistance phenotype and that some of these bacteria increase in frequency in resistant and selected mosquitoes.

## Results

### Origin of the strains

A highly resistant *An. coluzzii* colony collected from Banfora [22], Burkina Faso unexpectedly and rapidly lost resistance to pyrethroid insecticides 4 years after establishment as a lab colony,

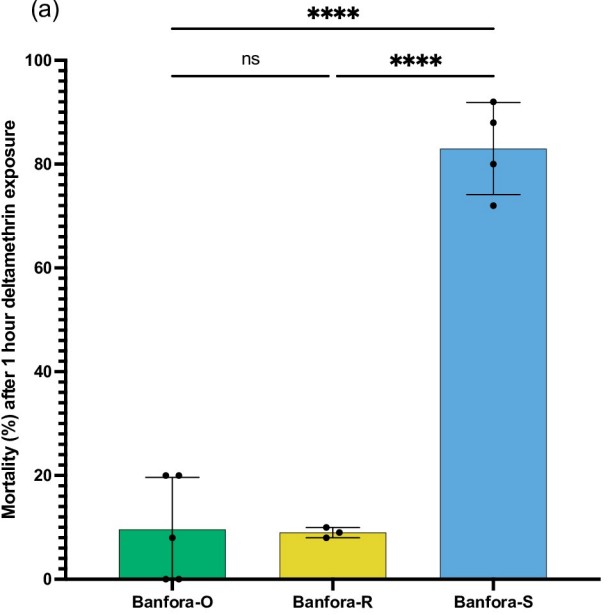

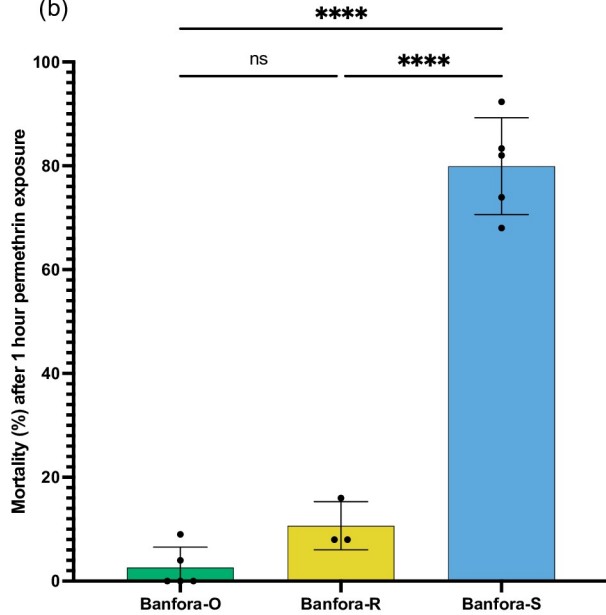

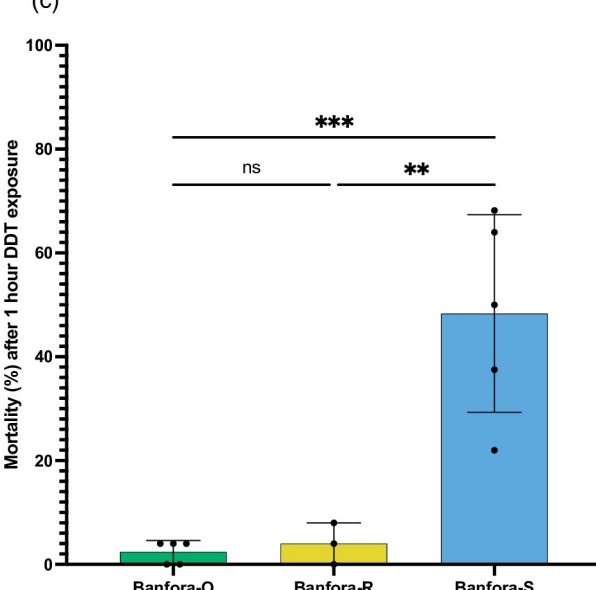

**Fig 1. Phenotyping of three Banfora lines.** 24-hour mortality after standard WHO assays for (a) 0.05% deltamethrin, (b) 0.75% permethrin and (c) 4% DDT. The Banfora-O population mortality is taken from phenotyping 6-months before loss of resistance, both Banfora-S and Banfora-R colony phenotypes are shown. Significant difference calculated by ANOVA followed by Tukey's *ad hoc* test. **** $p < 0.0001$ and ** $p < 0.005$.

despite regular selection with the pyrethroid deltamethrin (Fig 1). Resistance was restored to pre-loss levels after exposing a subset of the susceptible colony to 3 sequential rounds of delta-methrin selection (Fig 1). The temporary loss followed by rapid re-selection of resistance provided an opportunity to identify the mechanisms responsible for pyrethroid resistance in this strain. Throughout the study, Banfora-O will be used to reference the original resistant colony, Banfora-R the new re-selected colony and Banfora-S the susceptible colony.

## The restoration of pyrethroid resistance is associated with higher respiration rates

RNAseq analysis was carried out on four biological replicates from Banfora-O, Banfora-R and Banfora-S populations; Principal Component Analysis (PCA) showed that much of the variance was driven by Banfora-O compared to the two sister lines (39%) whilst PC2 separated Banfora-R and Banfora-S (17%) (S1 Fig). The closer relationship between Banfora-R and Banfora-S is not unexpected, given their separation of just four generations. To rule out contamination of the colony, WGS was performed and discussed below. Significantly down-regulated genes found in both Banfora-O and Banfora-R lines when compared to Banfora-S are enriched in transmembrane and ion transport (p $_{fisher's\ exact\ test}$ = 9.95e-15, 5.35e-9) and regulation of intracellular PH (p $_{fisher's\ exact\ test}$ = 3.17e-5). The up-regulated terms across both populations are highly enriched for NADH dehydrogenase activity (p $_{fisher's\ exact\ test}$ = 1.22e-11), oxidative phosphorylation (p $_{fisher's\ exact\ test}$ = 3.6e-13), cellular respiration (p $_{fisher's\ exact\ test}$ = 3.56e-11), mitochondrial membrane protein complex (p $_{fisher's\ exact\ test}$ = 7.74e-31) and respirasome (p $_{fisher's\ exact\ test}$ = 1.41e-28) suggestive of large changes to basal metabolism (S1 Table). To test this hypothesis, the respiratory rate of Banfora-S and Banfora-R lines was measured daily in adult females from age 3 to 7 days. At each time point, the resistant mosquitoes respired at a significantly higher rate than the susceptible counterparts, even when normalised for size (Fig 2A). Further, the resistant mosquitoes are significantly smaller than the susceptible (p $_{t-test}$ = 3.9e-3) with a mean wing length of 2.76mm compared to 2.85mm (Fig 2B), indicating body size is related to biological changes resulting in resistance.

Increased rates of respiration are linked with increased oxidative stress [29]. Previous work has shown that silencing a key oxidative stress sensing pathway, *MafS-Nrf-cnc*, is associated with a loss of pyrethroid resistance [24]. In addition to linking the pathway phenotypically with resistance, the study also produced a microarray data set characterising genes controlled by this pathway [24]. Comparisons of the genes regulated by the *MafS-Nrf-cnc* pathway with those differentially expressed between Banfora-R and Banfora-S reveals a high degree of overlap. Of the 428 significantly over-expressed and 359 down-regulated genes in this study which also present on the microarray chip, 214 and 117 are also regulated by the *MafS-Nrf-cnc* pathway respectively, a significant enrichment (p $_{hypergeometric\ test}$ < 0.0001). Further, the majority of genes show opposing expression patterns in the resistant lines and the *MafS-Nrf-cnc* pathway knockdown (83.2% of the up-regulated and 91.5% of the down-regulated genes) thus indicating that the Banfora-R population has higher expression of genes resulting in elevated levels of oxidative stress, possibly resulting from, or leading to, elevated respiration rates.

A recent study demonstrated large reductions in the oxidative phosphorylation pathway from 4-hours post-pyrethroid exposure [15]. To determine whether this phenotype is seen in the Banfora strain, and to further link respiratory rate to insecticide resistance, Banfora-R was exposed to a pyrethroid impregnated bed net and assayed for respiratory rate after 4-hours. A significant reduction in respiratory rate was seen post-exposure, indicating that pyrethroid resistance may require significant metabolic plasticity (Fig 2C).

## Evidence for the involvement of metabolic resistance

In addition to enrichment of gene families associated with respiration, up-regulated genes overlapping Banfora-O and Banfora-R are enriched in oxidoreductase activity (p $_{fisher's\ exact\ test}$ = 7.35e-4), precatalytic spliceosome (p $_{fisher's\ exact\ test}$ = 5.56e-6) and regulation of gene expression (p $_{fisher's\ exact\ test}$ = 2.1e-6) (S1 Table). The RNAseq data indicates that metabolic detoxification may be enabled by relatively few cytochrome p450s, which differs from previous transcriptomic studies implicating a wide range of these genes in resistance. Indeed, no

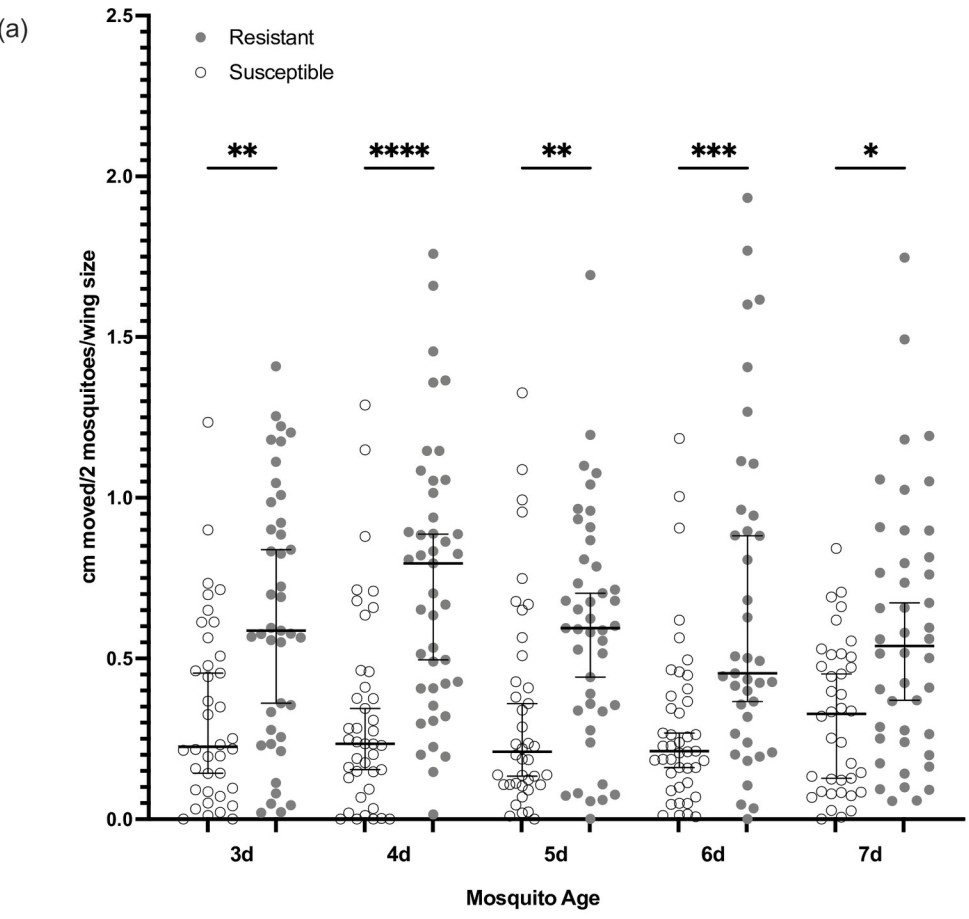

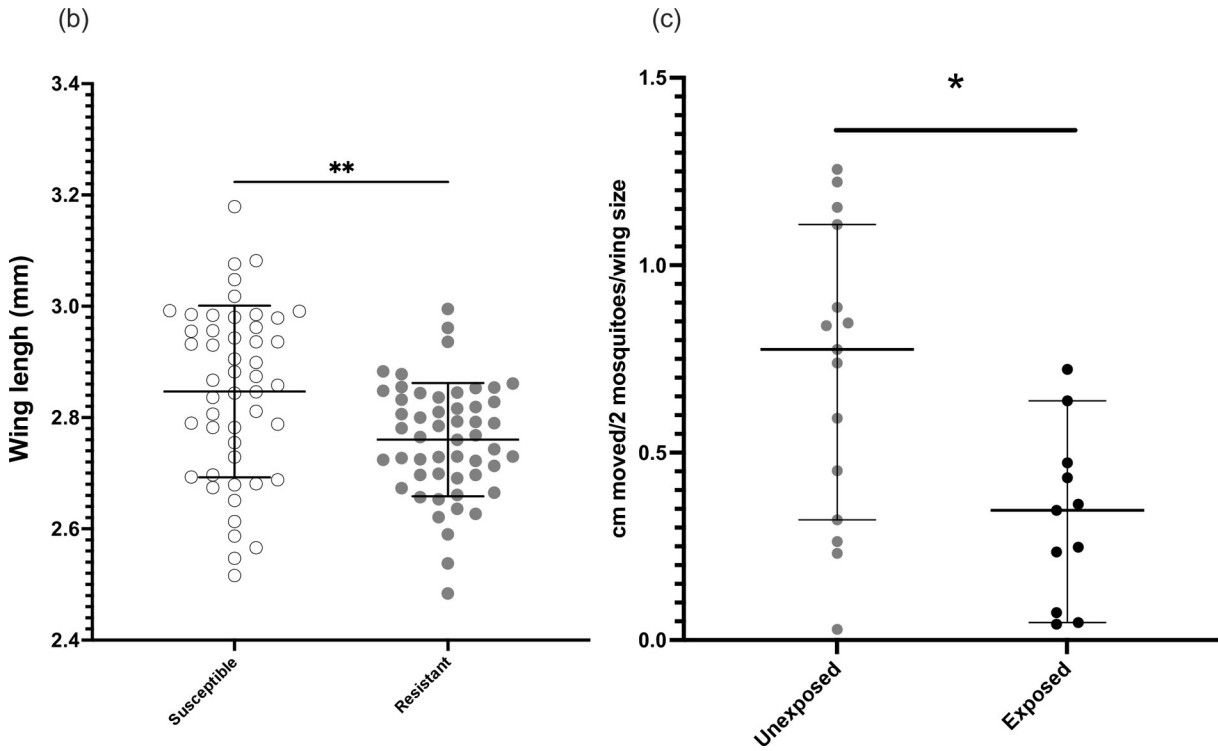

**Fig 2. Respirometer and wing measurements for resistant and susceptible lines.** (a) Centimetres moved per two mosquitoes as corrected for average wing size for each mosquito batch (y axis) acts as a proxy measure for the amount of $CO_2$ produced by the mosquitoes across the time points (x axis) for Banfora-R and Banfora-S. Significance calculated by a Kruskall-Wallis test. Error bars show median and 95% confidence intervals. (b) Wing size measurements for Banfora-R and Banfora-S mosquitoes (c) As in (a) comparing Banfora-R unexposed and exposed to the IG1 bed net. Error bars show mean and standard deviation. Significance calculated by t-test. **** $p < 0.0001$, *** $p < 0.001$, ** $p < 0.01$ and * $p < 0.05$.

enrichment of the 113 cytochrome p450s was seen; 8 are down-regulated in both Banfora-R and Banfora-O compared to the susceptible population, with just three P450s (*CYP6M2*, *CYP6P15P* and *CYP6AG1)*, upregulated in both resistant strains; of these, *CYP6M2* is a known pyrethroid metaboliser [19] and is likely contributing to the resistance phenotype. qPCR on 9 detoxification candidates shows the same pattern of minimal p450 involvement (S2 Fig). A small number of genes from other insecticide detoxification families are up-regulated in both Banfora-R and Banfora-O including *GSTS1*, previously linked to reactive oxygen species metabolism [30] (S1 Table). AGAP004690 and AGAP008449 were the highest differentially expressed genes in both resistant populations, both of which encode cuticular proteins; *CPF3* and *CPLCG5* respectively. In total, 33 genes were over 2-fold differentially expressed across both populations compared to the susceptible; in addition to the genes described above, these include the ABC transporter *ABCC8*, three trypsin genes and two NADH dehydrogenase sub-units. The two most down-regulated genes are AGAP002771 (a protein of unknown function) and AGAP011475 an envelysin (a metalloprotease). A total of 32 genes are down regulated by 50% or more in the resistant populations, including three cytochrome p450s, *CYP6Z1*, *CYP4C36* and *CYP304B1*.

## Inversion status but not gene duplications are linked with resistance on this strain

Whole genome sequencing of the three Banfora colonies revealed long divergent haplotypes and only modest inter-population differences (Fig 3A). There were few fixed differences between Banfora-S and Banfora-O, almost all on the X chromosome (S4 Table), indicating that contamination with a lab susceptible population is unlikely to have led to the loss in resistance. As with the RNAseq, PCA analysis showed that Banfora-S and Banfora-R displayed little differentiation and were more similar to each other than to Banfora-O, confirming the direct relationship of these two lines (Fig 3B). Post-filtering 6,928,092 SNPs were called and retained across the 96 individuals compared to PEST P4 (273,109,044 bases).

Chromosomal inversions are common within the *An. gambiae* complex [31] and using informative markers [32], inversions on chromosome 2 were seen in the Banfora colonies. The 2La inversion appeared fixed in the Banfora-O colony (n = 20) but was found at frequencies of 38% in Banfora S (n = 86) and 48% Banfora R (n = 86) (Fig 3C); the frequency of 2LA did not differ between Banfora-R and Banfora-S populations ($p_{\text{fisher's exact test}} = 0.187$). However, significant differences in the frequencies of the inversions on chromosome 2R were detected between the colonies. The 2Rb and 2Rc inversions are found at significantly higher frequency in Banfora-R (2Rb: 23%; 2Rc: 9%) ($p_{\text{fisher's exact test}} < 0.0001$; 0.0447) and Banfora-O (2Rb: 25%; 2Rc: 20%) ($p_{\text{fisher's exact test}} = 2e-4$; 0.0014) populations than Banfora-S (2Rb: 2%; 2Rc: 2%). Thus, among these well-known inversions, 2Rb and 2Rc track the loss and regain of resistance observed here.

A recent publication has linked gene duplication events with insecticide pressure [33]; however, the reversion of resistance in this population was not associated with copy number variation. We determined the frequency of the reported duplications surrounding detoxification genes in *An. coluzzii* [33] and found two in this population; Cyp6aap_Dup10 (covering four

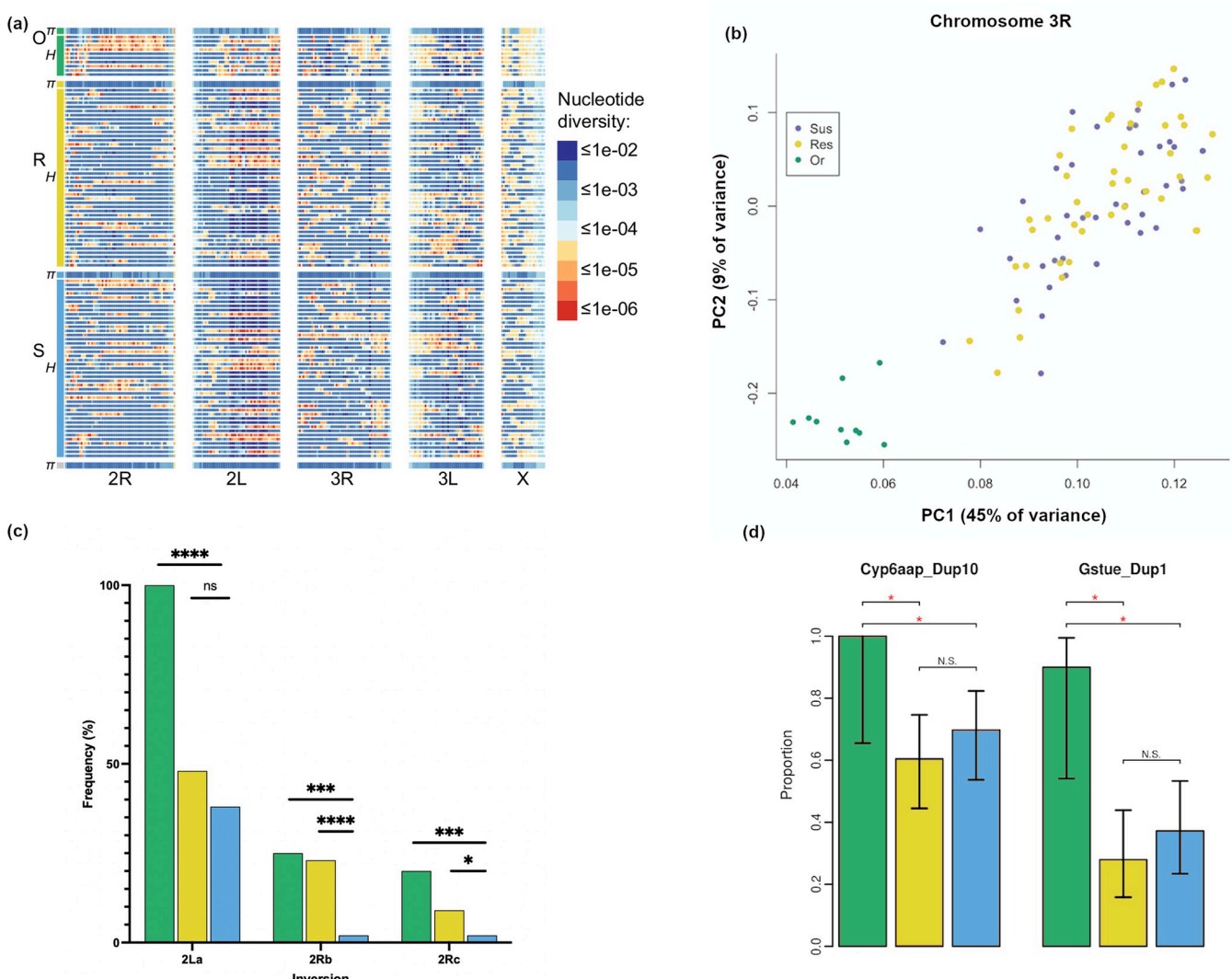

**Fig 3. Whole genome sequence results.** In each image Banfora-O is green and Banfora-R is yellow Banfora-S is blue. (a) Nucleotide diversity per population (π) and per individual (heterozygosity, H) along the length of each chromosome in 1 Mb windows. Diversity is similar among populations and overall (grey at bottom) but is heterogeneous along chromosomes suggestive of large haplotype blocks (b) PCA plot of chromosome 3R, chosen to represent the autosomal genome outside of large inversions (c) Frequency of inversion status in each population (d) Gene duplication scans showing the proportion of each population containing the two detected duplications. Significance calculated by Fisher's exact test. * = p < 0.05, *** p < 0.001, **** p < 0.0001, ns. indicates non-significance.

genes, *CYP6AA1*, *CYP6AA2*, *COEAE60* and *CYP6P15P*) and Gstue_Dup1 (containing *GSTE2* and a small portion of *GSTE4*). These duplications were found at high frequencies in the Banfora-O colony but at significantly lower frequencies in the Banfora-S and Banfora-R colonies (Fig 3D). RNAseq comparing Banfora-O to Banfora-S colony shows significantly increased expression of all genes within the duplications in the resistant population. In the Banfora-R line, just *CYP6P15P* is differential, indicating these duplications are putatively responsible for the increased transcript expression of the whole cluster (S2 Table).

To determine whether the increased respiratory rate was due to an increase in mitochondrial number, mitochondrial read counts were extracted and visualised across the length of the

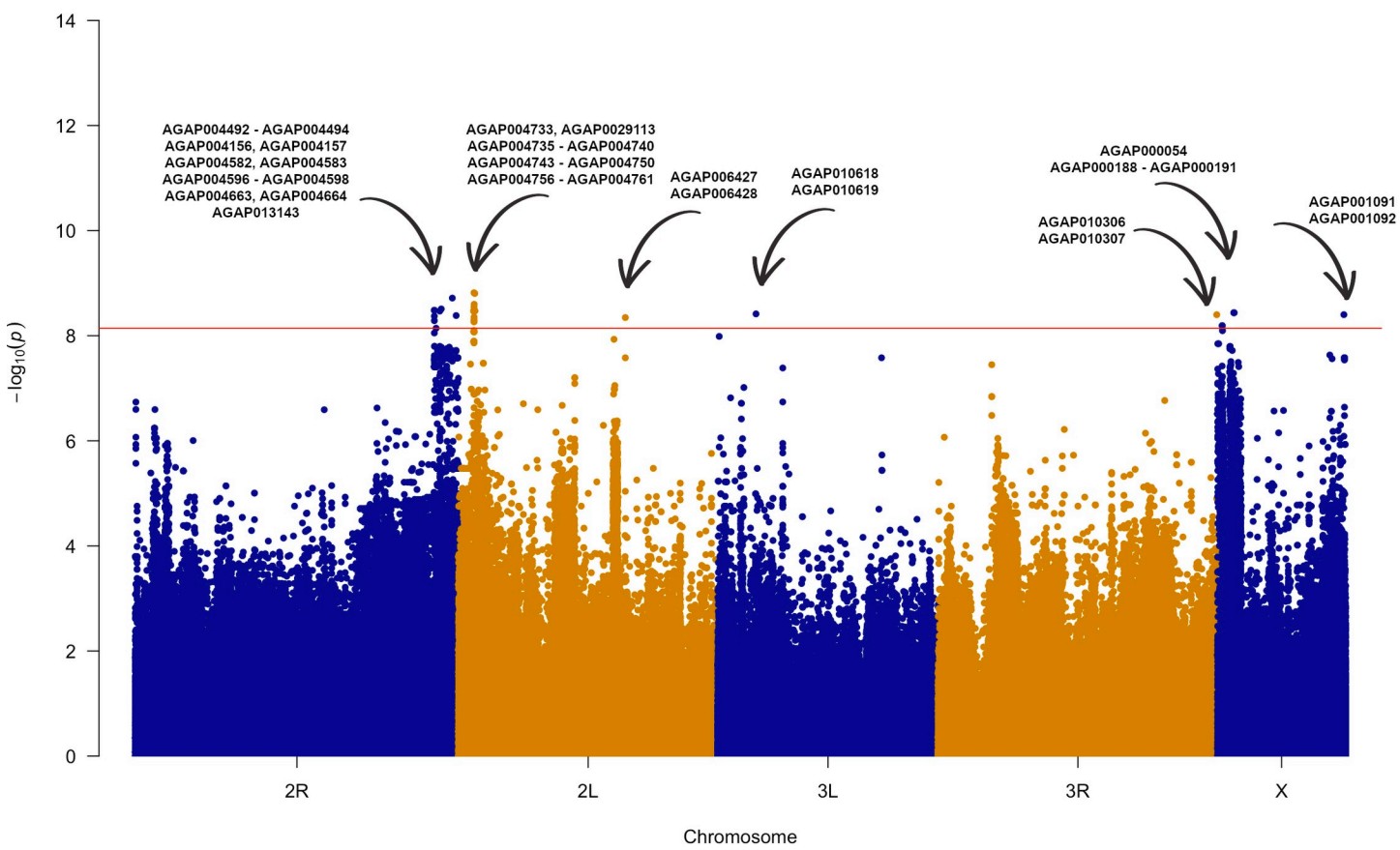

**Fig 4. Manhattan plot comparing reselected and susceptible populations.** GWAS-like analysis with populations as proxies for phenotype calculated using pyseer, likelihood ratio test p-values are plotted with the FDR p cut-off of p = 7.2e-9 shown with a red line. Alternative colours show the split between chromosomes. Each individual point is a SNP along each chromosome (x) and -log10 p value (y). Annotated genes are predicted from SNPEff, contiguous ranges are shown with '-'. Corrections for inversions and clonality were performed to account for population structuring.

genome. There was no difference in read depth between the susceptible and re-selected mosquitoes (S3 Fig) indicating that the increase in respiration not due to an overall greater number of mitochondria in the resistant populations.

## Genomic regions associated with restoration of resistance

Using a GWAS-like approach with populations as proxies for phenotype, we found 209 significant SNPs, of these 189 SNPs correspond to a large block on 2L ranging from 2920946 to 3085768, encompassing 28 genes (35 transcripts). SNPEff shows potential changes in 23 genes in this region (Fig 4, S3 Table), GO enrichments for these genes include mitochondrion ($p_{\text{fisher's exact test}}$ = 3.6e-2) and ribonuclease activity ($p_{\text{fisher's exact test}}$ = 5.99e-3). Nine non-synonymous changes associated with resistance were predicted with 8 in AGAP029113 and one in AGAP004735. Neither AGAP029113 nor AGAP004735 have assigned functions; however, the latter is a direct homolog of Meckel Syndrome, type 1 (Mks1) in *Drosophila* which is involved in cilium assembly. AGAP029113 has no direct homolog in *Drosophila* but has both homeodomain and SANT/Myb domains (IPR009057 and IPR001005) and a nuclear receptor co-repressor related NCOR (PTH13992) indicating that this gene has a regulatory function. STRING analysis of AGAP029113 predicts interactions with the ecdysone receptor, ultraspiricle and estrogen-related receptor indicating that this regulatory function may be related to hormonal

signalling. Neither AGAP029113 nor AGAP004735 are differentially expressed between the strains; however, the RNAseq analysis showed that a block of six contiguous genes in this region are all significantly up-regulated in Banfora-R (S3 Table) indicating either SNP-based changes or changes to transcriptional regulation driving the inheritance of this block.

In addition to this region on 2L we identified 20 additional SNPs classified as significant, one on 2L, a smaller block between 2R:56934652–56934669 containing 5 SNPs, five further SNPs at the end of the 2R chromosome, one SNP on each of 3L and 3R and finally seven SNPs found on the X chromosome. In each case, these SNPs are in intragenic regions, represent intron variants or cause up- or down-stream changes to genes following SNPEff terminology (S3 Table).

## Peaks of divergence

High $F_{ST}$ (>0.25) peaks with no fixed differences were seen across the chromosomes (Fig 5A, S4 Table), except for chromosome 3L, indicating regions of divergence between Banfora-R and Banfora-S. 21004 SNPs showed high $F_{ST}$, with the $F_{ST}$ peaks closely matching the SNP peaks seen in the GWAS-like analysis. Highly divergent SNPs appear in clear blocks and are shown in Fig 5A as blocks a-j; many of these SNPs show similarly high $F_{ST}$ in Banfora-O (Fig 5B, S4 Table) reinforcing their role in the resistance phenotype.

As seen with the GWAS approach, the centromeric regions on both 2R and 2L appear to be key in driving the resistance phenotype seen in these populations. The large block on 2R (19.3Mb, Fig 5 block 'b') comprises 1008 genes. Gene ontology (GO) enrichment analysis of genes present in this block shows significant enrichment for glutathione transferase activity (p $_{fisher's\ exact\ test}$ = 6.83e-4), oxidoreductase activity (p $_{fisher's\ exact\ test}$ = 3.32e-3) and glutathione metabolic process (p $_{fisher's\ exact\ test}$ = 3.1e-2). Given the high levels of respiration seen in this population, it may be that this region is buffering the excess ROS produced. 386 of the 917 genes detected by RNAseq in this region are differentially expressed, 213 of which are up regulated, including *GSTD1* and a number of heat shock proteins. The large block on 2L is 6.5Mb and overlaps the *kdr* locus (Fig 5, block 'd'), a gene known to increase resistance to pyrethroid insecticides [16]. Despite the overlap of this locus and the presence of 13 highly divergent SNPs within *kdr* (S4 Table), there is no significant difference in frequency of the classic *kdr* allele 995F (49% in Banfora-S and 64% in Banfora-R). The 1527T and 402L changes in the *kdr* locus have recently been linked with pyrethroid resistance [34] and are in perfect linkage in this population and mutually exclusive with the 995F mutation as previously reported [34]; again, there is no difference in frequency between strains. The remaining peaks illustrated in Fig 5A are described in Appendix 1.

## Microbial composition is associated with insecticide resistance

Significant differences in microbial composition were seen between the Banfora-R and Banfora-S lines, with clear relative increase of *Elizabethkingia anophelis* and *Herbaspirilum sp* (Fig 6). No differences in species richness were seen between the two groups and a Bray-Curtis dissimilarity shows high overlap of the Banfora-R and Banfora-S populations (S4 Fig); however, a significant difference in beta diversity (p $_{PERMANOVA}$ = 7.9e-4) is seen demonstrating differences in species compositions between the populations. To determine the highest contributions to microbiome weighting, operational taxonomic units (OTUs) were selected that added most to the between sample diversity. These included the endosymbionts *E. anophelis*, *Asaia borgorensis* and *Serratia* sp Ag1. Other bacteria significantly contributing to the microbiome include *Rhizobium tropici*, *Herbaspirilium* sp, *Ochrobactrum* sp, *Acinetobacter soli*, *Pantoea dispersa* and *Acetobacter* sp. Of these bacteria, *Pantoea* (p $_{Mann-Whitney}$ = 1.8e-3), *Acinetobacter*

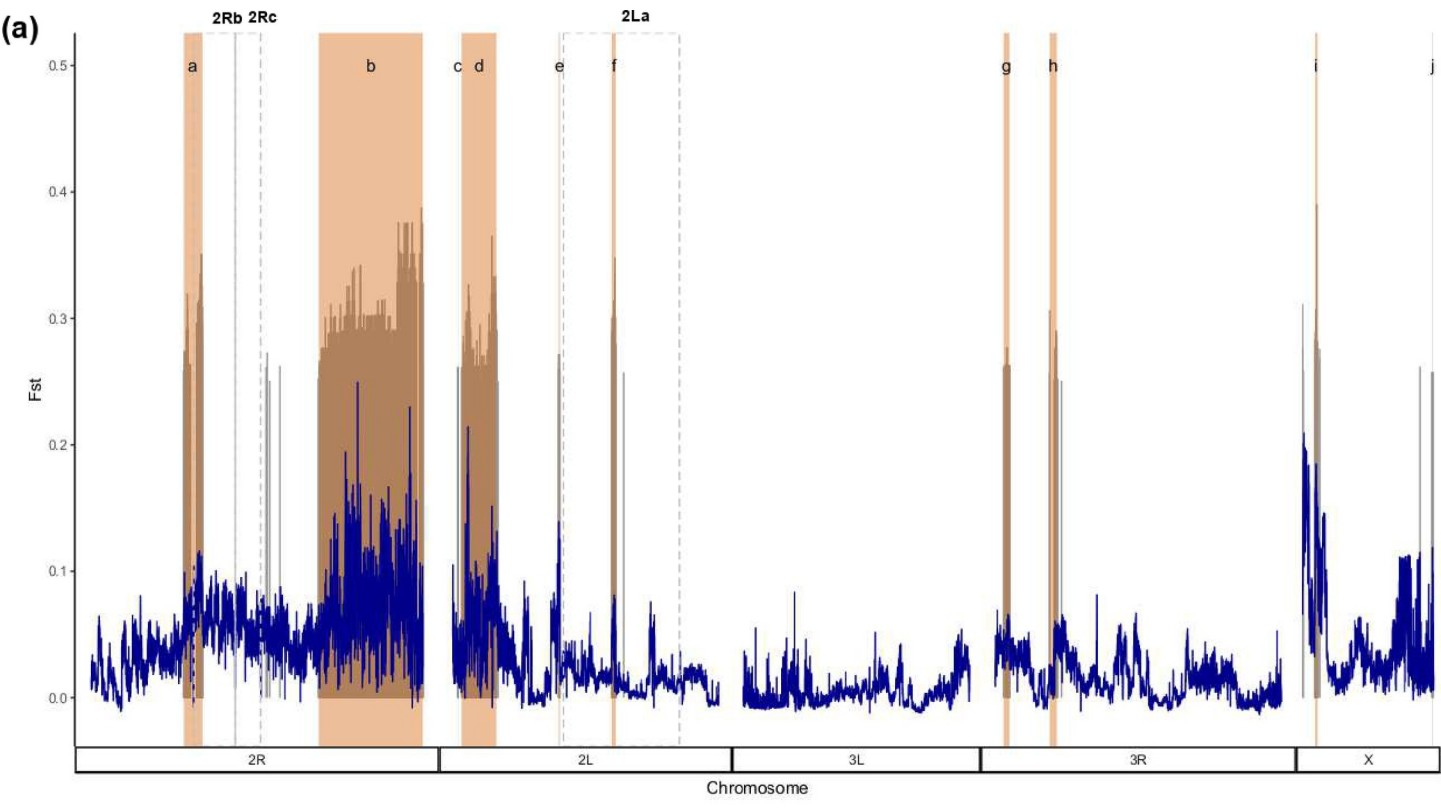

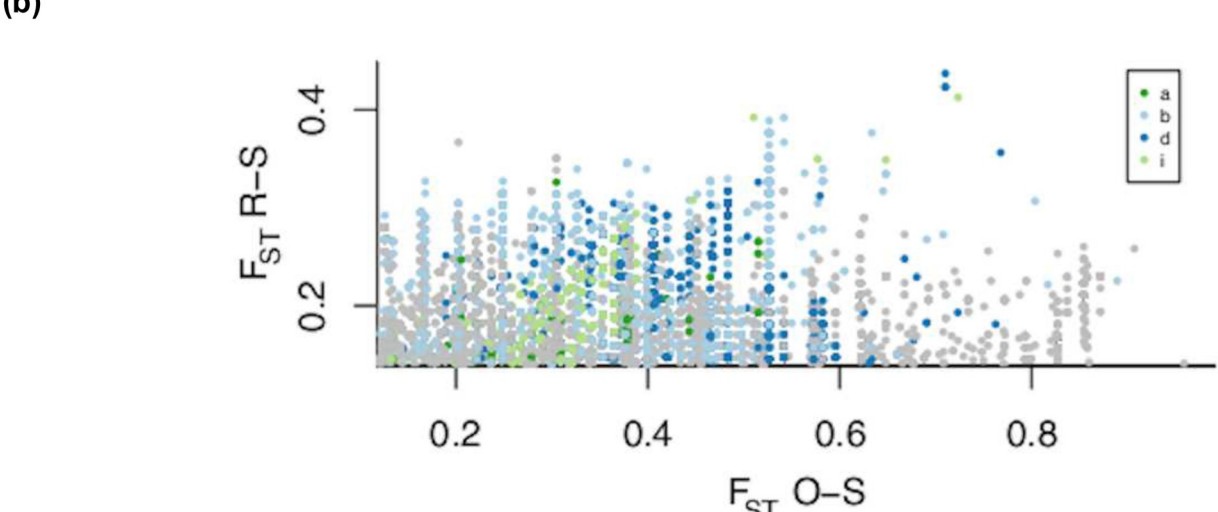

**Fig 5. $F_{ST}$ between Reselected and Susceptible populations.** (a) $F_{ST}$ (y axis) and chromosomal position (x axis) of reselected compared to susceptible populations. Dark blue shows the average $F_{ST}$ over 10kb windows and grey are individual SNPs with $F_{ST} > 0.25$. Shaded in pink are regions identified as blocks of divergent SNPs, the inversions are highlighted by dashed grey boxes. (b) SNPs showing high $F_{ST}$ and allele frequency difference in the same direction relative to Banfora-S, for Banfora-O and Banfora-R. Several SNP blocks as in part (a) (blocks a, b, d, and i) are highlighted.

*soli* (p Mann-Whitney = 7e-4) and *Serratia* (p Mann-Whitney = <1e-4) are at significantly reduced abundance in the Banfora-R population compared to Banfora-S, whereas *Elizabethkingia* (p Mann-Whitney = 1.11e-2), *Rhizobium* (p Mann-Whitney = <1e-4) and *Herbaspirilium* (p Mann-Whitney = 1.09e-2) are at significantly higher abundance in the Banfora-R population compared to the

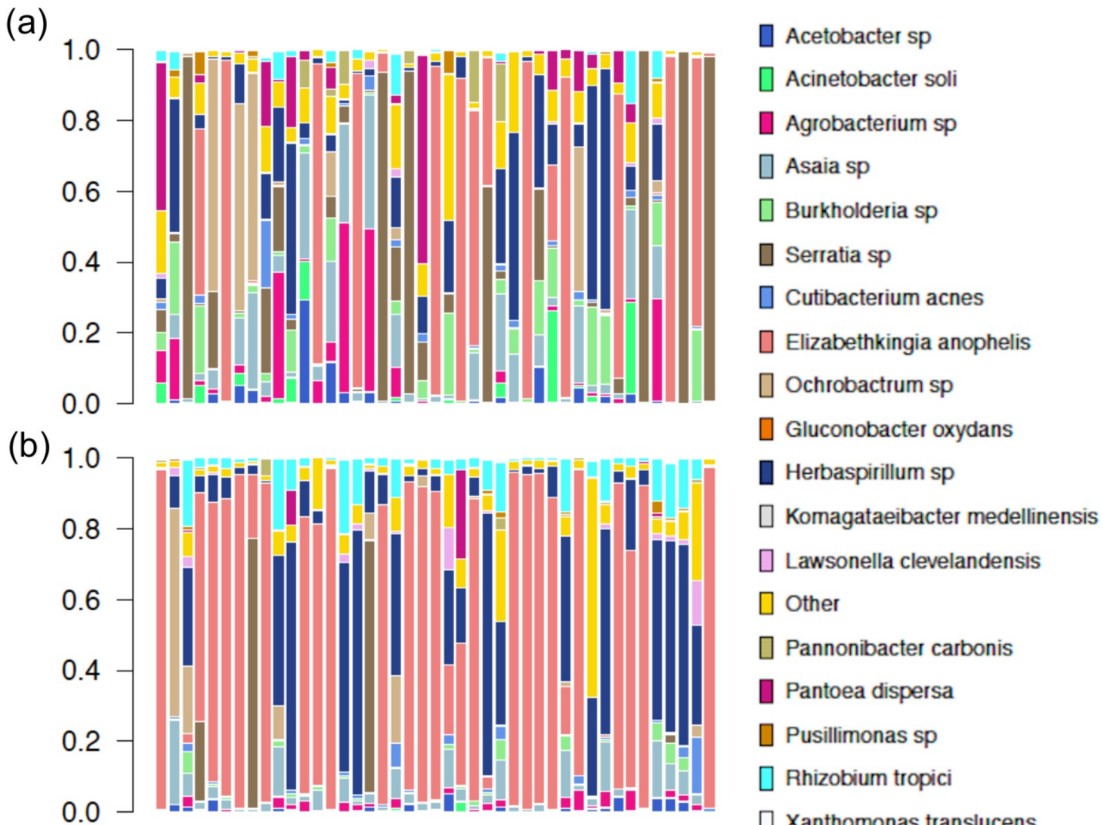

**Fig 6. Abundance plots.** Species abundance (y) for each biological sample (x) for (a) Banfora-S and (b) Banfora-R populations. Other represents the total sum of all other abundances within each individual.

Banfora-S. *Asaia*, *Ochrobactrum* and *Acetobacter* show no changes in abundance after selection and so are unlikely to contribute to resistance (Figs 6; S5).

To confirm the presence of the bacteria within the mosquito populations, PCR was performed on whole DNA extracts for the selected OTUs and positive bands sent for sequencing. Of the bacteria selected as greatest contributors to beta-diversity, only *Serratia* couldn't be confirmed by PCR, potentially indicating an unstable infection or inadequacy of the published primers for anopheline *Serratia*. Additionally, phylogenies were reconstructed from extracted 16S sequences for high abundance bacteria showing that these nest within samples isolated from mosquitoes (S6 Fig). To further explore the presence of these bacteria, ORFs were BLASTed against the non-redundant species database. Whole length genes were found directly attributable to the bacteria identified in the earlier analysis for each treatment group (S5 Table). In addition to the bacteria, 63 fungal reads were detected in Banfora-R which were absent in Banfora-S. Further, the Banfora-R population has a 316 amino acid match to the RNA virus Xincheng mosquito virus indicating a potential integration event. These data indicate a change in microbial composition after selection for resistance, potentially indicating that the microbiome is either directly contributing to the resistance phenotype or that the use of insecticides preferentially selects for certain bacterial species.

## Discussion

This study utilises multiple -omics data and phenotypic studies to explore causative factors of pyrethroid resistance in an *Anopheles coluzzii* colony from Burkina Faso, after a sudden and

dramatic loss of the phenotype. The subsequent re-selection of the susceptible population, and stored material from the original Banfora-O colony, present a rare opportunity to explore the causal mechanisms of pyrethroid resistance in populations from identical genetic backgrounds. Despite similarities between the two resistant populations, there are clear genomic and transcriptomic differences, underlining the importance of the two comparators in identifying changes necessary for resistance. The data here reveals a much more complex story than often reported in resistance research and shows that resistance is not entirely attributable to previously characterised mechanisms. Here, we show that the respiratory rate is elevated in resistant mosquitoes, indicative of large-scale changes in the mosquitoes' basal metabolism. Further, we highlight a clear association of resistance with divergent regions of the genome. Finally, we demonstrate a change in microbial composition upon re-selection for pyrethroid resistance.

Loss of resistance in this population was associated with a strong reduction in expression of genes involved in the oxidative phosphorylation pathway, which was subsequently restored upon re-selection. The change in expression of this pathway is closely mirrored phenotypically by changes in respiration rates, with resistant mosquitoes having higher levels of respiration. Due to these changes, higher resistance levels are likely to incur a high fitness cost and may account for the lack of stability of resistance in this strain. Fitness costs associated with the gain of resistance in wild populations are well documented throughout the *Insecta* class; however, long term selection pressure with single insecticides has been shown to lead to a loss of these associated costs [35, 36]. Indeed, the Banfora population is atypical in the colonised pyrethroid resistant Anophelines, where resistance persists over many generations in the absence of selection pressures [12]. Within this population, changes in respiratory rate may result in the smaller body size through depleted energetic stores [37]. Although previous studies indicate that larger body size increases vigour and hence resistance, several studies have shown that insecticide resistance and exposure leads to smaller body sizes [38–40], mirroring the results of this study. Remarkably, the changes seen to the oxidative phosphorylation pathway closely mirror but oppose those seen after exposure to pyrethroid insecticide in a different resistant population from Burkina Faso [15]. Interestingly, exposure to pyrethroid insecticides also causes a significant reduction in respiratory rate in the resistant Banfora population, and as pyrethroids are known to cause oxidative stress [41], this further implicates metabolic plasticity, potentially through modulation of oxidative stress, in pyrethroid resistance and response.

A higher basal metabolic rate is likely to result in higher levels of oxidative stress. Whilst oxidative stress levels were not directly measured in this study, a putative link between the elevated basal metabolism and oxidative stress signalling was identified by comparison of the transcriptomic changes seen between Banfora-S and Banfora-R and those identified in a previous study perturbing the oxidative stress signalling pathway via silencing a component of the *Maf-S-cnc* pathway. Previous work has shown that perturbation of this pathway leads to increased mortality post-exposure to pyrethroid insecticides [24]. There is a clear correlation between genes differentially expressed in this study and those perturbed by disruption of *Maf-S* signalling. Further, this signal displays clear negative reciprocal overlap, as expected if higher basal metabolic rate is causing increased oxidative stress. Taken together, these data indicate a role for oxidative stress in the resistance phenotype within this population.

Despite the clear evidence of the key role detoxification genes play in the metabolic breakdown of insecticides in anopheline populations [9, 42], there is little evidence that this Banfora laboratory population relies on these gene families in aggregate to confer resistance. Only three cytochrome p450s are overexpressed in both the original and reselected populations, *CYP6M2*, *CYP6P15P* and *CYP6AG1*. *CYP6M2* is a well-studied pyrethroid metaboliser and

hence may be contributing to the resistance phenotype [19] but the latter two p450s have not been studied despite *CYP6P15P* being present in a gene duplication seen in wild populations [33]. There is some evidence for cuticular resistance in this population, as CPLCG5 is the most up-regulated gene across both datasets and this gene has previously been shown to impact pyrethroid resistance in *Culex* mosquitoes [43]. Interestingly, *CYP6Z1* which has previously been implicated in DDT resistance [44] is downregulated in these populations despite high levels of resistance to this chemistry.

The transcriptional changes overlapping Banfora-R and Banfora-O are small compared to the long gene lists seen in prior studies in which a resistant population is compared to a genetically distinct susceptible population [14]. Indeed, just 3.5% of the annotated genes are up-regulated, whilst 2.8% are downregulated and yet strong enrichments are present in these data; this indicates that the resistance phenotype within this population modifies expression of entire pathways rather than reliance on up-regulation of single detoxification genes.

Whole genome sequencing on individual females of each population reveals clear divergence between the Banfora-R and Banfora-S populations. Interestingly, the well characterised inversions present on chromosome 2 of the *An. coluzzii* genome show significant differences in frequency between the three populations. One striking region is the 2Rb inversion which partially overlaps SNP block 'a', being present at significantly higher frequency in both the Banfora-O and Banfora-R populations than the susceptible, implicating this region in resistance in this population. Other than the larger and better studied 2La inversion, 2Rb is the only other inversion found widely across sub-Saharan Africa in multiple anopheline species [31]. The 2Rb inversion has been linked with host preference in *Anopheles arabiensis* [45] and larval breeding habitat [46] and desiccation tolerance [47] in the *An. gambiae* complex but thus far has not been linked to resistance. Other SNP blocks show a similar degree of divergence including 'b', 'd', and 'i' highlighting multiple loci playing putative roles in resistance in this population.

WGS also reveals two previously described copy number variations [33] in this population which are at higher frequency in Banfora-O than either Banfora-R or Banfora-S colonies, indicating that they are not necessary for resistance in this population. Despite revealing both highly divergent $F_{ST}$ peaks and SNPs significant via GWAS methodology, there is not enough resolution in this dataset to identify individual SNPs with a role in pyrethroid resistance, likely due to the high linkage disequilibrium in these captive populations. However, clear divergent blocks show an association with pyrethroid resistance, with high $F_{ST}$ in the Banfora-R and Banfora-O colonies. Interestingly, one such block overlaps the *kdr* locus but there is no association with known causal SNPs in this locus and resistance, indicating that the haplotype block may be related to a different gene in this region. The majority of the SNPs are found in noncoding regions and so may play a role in transcriptional regulation, but further studies will be needed to pinpoint the importance of these SNPs. Further, a region of 2R shows divergence between Banfora-R and Banfora-S and is enriched in genes involved in glutathione reductase activity. The glutathione pool is a redox buffer found within cells and is often used as a proxy for oxidative stress [48]; this may indicate that genes involved in reducing the oxidated redox pool help maintain redox levels which are increased due to increased respiration.

Whole genome sequencing of *An. funestus* populations has previously identified potential markers for pyrethroid resistant, including mutations in GSTE2 [49] and the promoter region of CYP6P9a [50]. Despite these promising steps, our results highlight a major challenge for rapid molecular detection of insecticide resistance in *An. coluzzii*: that it may not depend on one or a few known large-effect loci, but rather a complex causal architecture involving whole metabolic pathways shaped by many genes as well as nongenetic factors. Thus, future surveillance toolkits may need to transcend the candidate gene approach (e.g. Donnelly et al. 2016

[51]) and incorporate numerous polymorphisms, perhaps alongside gene expression data, microbiome markers, or other indicators.

An association between the microbial composition and resistance was seen in these samples, something previously noted in resistant populations [25, 27, 52]. However, even though the colonies were maintained in the same insectary by the same technician, we cannot rule out stochastic changes due to bottlenecking of the colony, which requires further exploration. Nevertheless, we show a clear increase in abundance of the known commensal *Elizabethkingia* in Banfora-R, which has not previously been linked to resistance. Of the other OTUs, *Acinetobacter* reduction in wild *Anopheles albimanus* has previously been reported in fenitrothion resistant mosquitoes [27], and *Pantoea* reductions have been linked with pyrethroid exposure [25] in agreement with results here. *Rhizobium*, a nitrogen fixing bacteria traditionally associated with plants has not previously been shown to be present in the mosquito microbiome but given the differential abundance within on extraction round, it is unlikely to be a reagent contaminant. *Herbaspirillum* and *Ochrobactrum* have previously been reported in the microbiome [53], but the former has not before been linked to resistance, as seen here. Strikingly, *Serratia* abundance is dramatically reduced in Banfora-R compared to Banfora-S and is in agreement with a recent discovery in Côte D'Ivoire showing that this bacteria is strongly associated with pyrethroid susceptibility in *An. coluzzii* [26]. Further study is needed to determine whether any of these individual bacteria contribute to the insecticide resistance phenotype, or conversely if any of these microbiome changes reliably occur in response to metabolic regulation underlying resistance and could serve as indicators for it.

This study provided a unique opportunity to compare two resistant populations and a susceptible population from the same genetic background, removing the confounding factor of the differences in genetic background of lab adapted susceptible populations. Here, we show evidence for involvement of relatively few metabolic detoxification genes. In addition, an increased respiratory rate appears to directly contribute to pyrethroid resistance through up-regulation of the oxidative phosphorylation pathway. Further, clear genetic signatures associated with resistance are seen, including an association with the 2Rb inversion and several blocks dispersed across the genome. Finally, we demonstrated a change in the microbial profile in resistant mosquitoes, further emphasising the need to study the impact of the microbiome. Overall, this study clearly demonstrates a hitherto underappreciated range of resistance-related changes, including changes to the genome and microbiome, that challenges prospects for simple DNA based diagnostics for resistance.

## Methods

### Mosquito rearing conditions

Mosquitoes were reared under standard insectary conditions at 27˚C and 70–80% humidity under a 12:12 h photoperiod. The *An. coluzzii* colonies used in these experiments were derived from the Banfora strain from the Cascades District of Burkina Faso. The Banfora colony is resistant to pyrethroids and DDT and was maintained under deltamethrin selection pressure in the insectaries at the Liverpool School of Tropical Medicine since 2014 [12]. In September 2018 after routine phenotyping, it was noticed that Banfora had significantly higher mortality after exposure to pyrethroid insecticides, prior to this full resistance was seen in March 2018. This provided the opportunity to generate two lines from the same parental population, Banfora-S which had lost much of its resistance and reselected Banfora-R line. The Banfora-R line was generated by exposing 3 consecutive generations to 0.05% deltamethrin WHO tube papers for between 30 minutes and 2 hours (see Fig 7).

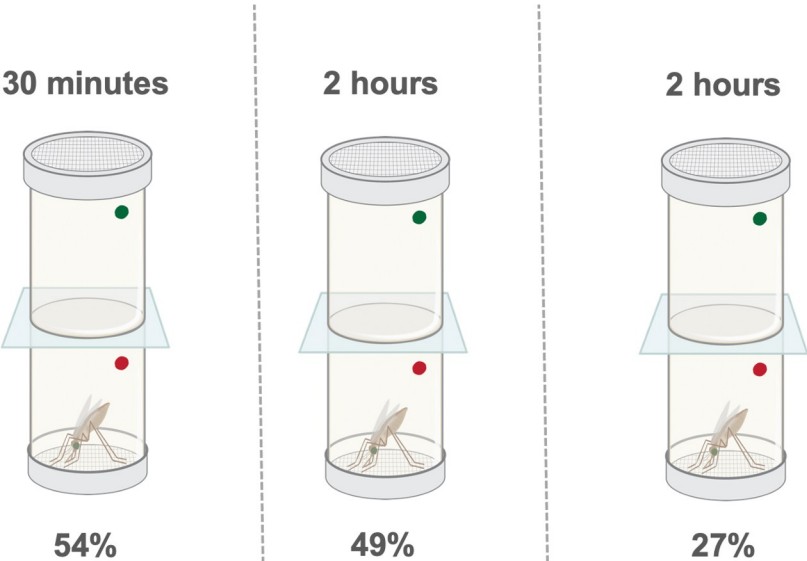

**Fig 7. Selection regime for the susceptible colony.** WHO deltamethrin tube selection regime implemented to reselect Banfora S to full resistance. The generations are shown separated by a dashed line. The durations of each exposure to 0.05% deltamethrin are given above the tube, with the mean mortality for each generation below. Mortality for the first two selections are calculated based on entire colony selection (N = 409; N = 514), whilst the final calculation of mortality represents a subset of the selected cage (N = 115).

## Bioassays

WHO diagnostic bioassays were performed for each population using WHO tube assays with 0.05% Deltamethrin, 0.75% permethrin and 4% DDT [54]. A minimum of 3 biological replicates were used, with 25–30 treated females present in each tube. For each assay 20–25 female mosquitoes were simultaneously exposed to untreated papers as a control. Post-exposure, mosquitoes were left in a control tube, under insectary conditions for 24 h, with 10% sucrose solution and mortality recorded. Analysis of mortality data was done on normal data using an ANOVA test followed by a Tukey post hoc test. Graphs were produced using GraphPad Prism 7. For exposure to the alpha-cypermethrin containing IG1 bed nets, an exposure using a cone test was used as previously described [12], mosquitoes were then left to recover for 4 hours before being placed in a respirometer with similarly treated unexposed mosquitoes as a control.

## RNA Extraction and analysis

At 10am, 3–5 day old presumed-mated adult females were snap frozen in the -80˚C, 5 individuals were used for each of the 4 biological replicates, for the RNAseq. RNA was extracted using a Picopure kit (Thermo) after homogenisation with a motorised pestle as previously described. Quality and quantity of the RNA was then analysed using an Agilent TapeStation and Nanodrop respectively and sent for sequencing at Centre for Genomics, Liverpool, UK. The fastq files were aligned to PEST 4.2 using Hisat2 [55] and then counts extracted using featureCounts [56] using default parameters; PEST 4.2 [57, 58] fasta and GFF files are available from Vector-Base (vectorbase.org) [59]. Differential expression analysis was performed using DEseq2 v3.10 in R [60], following the standard protocol. Briefly, count data was read in from the feature-Counts output, sample metadata including sample names and treatment were passed to DESeqDataSetFromMatrix, variance stabilised dispersions were then calculated and PCA

performed on this dataset. Genes with an average of less than 10 reads per sample were removed. Differential expression was then carried out using DESeq and lfcShrink from apeGLM v3.10 [61], following DEseq2 instructions. Significance was taken as adjusted $p \leq 0.05$.

### Detoxification family TaqMan

RNA was extracted as above in triplicate from a different generation of 3–5-day old female mosquitoes from each population. One to four micrograms of RNA were then reverse transcribed using OligoDT and Superscript III (Thermo) as previously described. The resulting cDNA was cleaned using a Qiagen PCR Purification column (Qiagen) and quantified; cDNA was subsequently diluted to 4ng/μl as a template for qPCR. Primers, probes and multiplex combinations used in this reaction were as previously described [12, 62]. PrimeTime Gene Expression Master Mix (IDT) was used with primers and probes at a final 10μM in 10μl. The qPCR reaction was carried out on an MxPro 3005P with the following conditions 3 min at 95°C followed by 40 cycles of 15 s at 95°C; 1 min at 60°C. Ct values were exported and analysed using the ΔΔct methodology, using RPS7 as an endogenous control and compared to the Banfora-S population.

### Respirometer

To determine the respiration rate of resistant and susceptible Banfora populations, two individual female mosquitoes were placed in one tube following previous published methodology [63]. Briefly, a 1000uL pipette tip, which had been cut and glued to a glass micropipette was placed into a tip holder over a clear container filled with dyed water. Each pipette tip contained soda lime between two pads of cotton wool. The mosquitoes were knocked down on ice and added to each tip before covering with clay. The mosquitoes were left to recover for fifteen minutes the assay began. Each respirometer allowed for a total of 12 tubes, with one control empty tube for each treatment group. The respirometer was performed in triplicate using different generations of mosquitoes. Images were taken of the water level immediately after mosquitoes were aspirated into the tubes and a second images was taken 45 minutes later before aspiration of the mosquitoes back into cups where they were maintained on 10% sugar under insectary conditions. Images were taken using a camera clamped into a stand. ImageJ was used to quantify the distance the water moved as a proxy for respiration, negative control movement was accounted for by simple subtraction. Any negative values were assumed to be 0. Respirometry data were adjusted to account for variations in mosquito size. Wing lengths of 15 randomly selected female mosquitoes, taken from the same colony cage, on the same date, were determined following previously published protocols. Each individual biological replicate was corrected using an average of the 15 mosquitoes taken from the same cage.

### Comparison with Maf-S knockdown

To compare the genes differential in the current RNAseq dataset and those in the Maf-S knockdown array, the full array data was used from Ingham et al. 2017 [24] and merged with the overlapping significant genes from the RNAseq. The fold changes for each experiment were then extracted and counted as (i) opposing and (ii) overlapping directionality.

### Whole genome sequencing

DNA was extracted from single female mosquitoes for 43 Banfora-R mosquitoes, 43 Banfora-S mosquitoes and 10 Banfora-O mosquitoes using a Qiagen DNeasy Blood & Tissue Kit

(Qiagen) following manufacturer's instructions. Whole genomic DNA was then sequenced with 151 bp paired-end reads on an Illumina HiSeq X instrument at the Broad Institute, using Nextera low-input sequencing libraries. Reads were aligned to the *Anopheles gambiae* PEST reference genome (assembly AgamP4 [57, 58]) using bwamem v. 0.7.17 [64] (command: bwa mem -M) and samtools v .1.8 [65] (commands: samtools view -h -F 4 -b, samtools sort, samtools index). Variants were called using GATK v. 3.8–1 [66], using hard filtering of SNPs with QD < 5 and/or FS > 60, and indels with QD < 2 and/or FS > 200 (—max-gaussians 4). Mitochondrial read depth was extracted for each sample using samtools [65] view for 'Mt' and coverage pulled using samtools depth function for combined and sorted BAM files for each of the susceptible and reselected colonies.

## Inversion status

Inversion status was determined by extracting previously published tagging SNPs for each inversion known to occur in the *Anopheles gambiae* species complex [32]. These regions were then extracted using vcftools [67].

## Population genomic analysis

Whilst the assumptions of $F_{ST}$ are a better match for a structured population dataset like this, a GWAS pipeline provides a natural threshold for identifying meaningful SNPs and so both were explored in this study. The vcf file was filtered using vcftools v0.1.17 [67] with minimum depth of 5, minimum quality score of 20, Hardy Weinberg equilibrium of 1e-6 and minor allele frequency of 0.01. BCFtools [65] was then used to remove SNPs with high missingness (> = 0.05) resulting in 6,928,092 variants and a 0.996 genotyping rate. For production of PCA plots, BCFtools was used to prune SNPs with ld >0.8 in 10kb windows. PCA plots were produced using the—pca flag in plink v2 [68], and corresponding eigenvectors read into R and plotted with ggplot2 [69]. Plink was used for $F_{ST}$ analysis, with sex specified as female and Banfora-S coded as '0' and Banfora-R or Banfora-O as '1' using the–fst case-control flag. SNPs with inflated $F_{ST}$ due to missing calls were removed. $F_{ST}$ was then plotted using the ggplot2 R package. SNPeff v4.3 [70] was used to define the effect of each SNP within the vcf.

To calculate heterozygosity and π, we retained sites for which all individuals showed coverage between 8 and 50 x with no missing data, excluding sites with ExcessHet > 30, and we examined nucleotide diversity in 1Mb blocks across the genome. Because of filtering, this approach likely underestimates true diversity but accurately portrays relative diversity among individuals and among genomic regions.

To account for the clonal nature of the individuals used in this study, Pyseer v1.3.1 [71] was used to account for strong confounding population structure. A whole genome phylogeny was generated using SNPhylo v20180901 [72], with the apeR v5.3 [73] package being used to extract phylogenetic distances between individuals. Due to the high number of variants, the vcf file was randomly thinned using the–thin 1000 flag on vcftools, resulting in 197928 sites being retained. The large inversion present on the *Anopheles* 2L chromosome and the differing inversion statuses of the individuals demonstrated high level structuring on the PCA plots and so were passed to Pyseer as a covariate file. Finally, a phenotype file was generated, with Banfora-S coded as '0' and Banfora-R as '1'. Pyseer was then ran with the–vcf,–phenotypes,—covariates and–distances flags with the files generated as described. Manhattan plots were produced on R using the Manhattan function in the qqman package. An lrt p $\leq$ 7.2e-9 was used for significance to correct for false discovery rate. All commands were sped up through the use of the Parallel package [74].

## Genome duplication scans

CNV detection was performed as described in Lucas et al 2019 [33], focusing on the five regions with previously identified CNVs of interest (Ace1, Cyp6aa / Cyp6p, Cyp6m / Cyp6z, Gste, Cyp9k1). Briefly, CNV alleles previously identified from Ag1000G data were detected from their associated discordant reads and soft-clipped reads. The possibility of novel CNVs in this sample set was investigated by applying a hidden markov model through normalised coverage data calculated in 300bp windows and also by visualising the coverage data across the regions of interest; this revealed no previously unknown CNVs.

## Enrichment analysis

All enrichment analysis was performed on VectorBase release 53 using built in GO, KEGG and Metacyc enrichments for *An. gambiae* PEST4.2. Benjamini-Hochberg corrected p values are used throughout, with significance $p \leq 0.05$.

## Microbiome reads

To determine the presence of bacterial reads in the BAM files, unmapped reads were pulled from the bwa mem alignment BAMs using the samtools view -b -f 4 command and converted to fastq files using bedtools bamtofastq command. Each step was aided through the use of Parallel [74]. The latest Centrifuge v1 [75] database was pulled from https://github.com/rrwick/Metagenomics-Index-Correction following the publication on improved databases [76]. Centrifuge v1.0.4 [75] was then run for each individual mosquito and converted to a kraken output using the -krereport function. Kraken reports were then visualised using Pavian [77]. To ensure adequate read depth, bacteria had to contain over 500 reads in at least 5% of the 96 samples. Bacteria were further filtered by abundance values of >0.01 in at least 5% of the samples. Kraken reports filtered as stated above were analysed following (https://rpubs.com/dillmcfarlan/R_microbiotaSOP) with the vegan v2.5.6 and SpadeR v0.1.1 packages in R (CRAN). All permutation tests had 10000 permutations. A Kruskall-Wallis test was used to compare alpha-diversity, a PERMANOVA for beta-diversity and a Mann-Whitney for comparing relative abundances. Data display was achieved using ggplot2.

To confirm the results from the Centrifuge database, contigs were assembled for each population by combining individuals within the differing population using megahit v1.2.9 [78]. The contigs were then converted into 6-way open reading frames using TransDecoder.LongOrfs. The orf were then BLASTed against PEST 4.2 and *Anopheles* reads removed. The remaining protein reads were then BLASTed against an NCBI non redundant protein database, identifying only the top hit and pulling the taxon id. An R script was then written using NCBI taxon dump to identify whether the orf relates to 'virus', 'bacteria' 'fungi' or 'other' (https://github.com/VictoriaIngham/Banfora_Paper). The longest read from each bacterium of interest was BLASTed against the NCBI database to identify the appropriate genome assembly to align to. Bacterial genomes were assembled by aligning to the reference genome using Hisat2 [55] for each of the major bacterial species with an average read depth of $> 50$ reads. The number of reads aligned in each BAM file was then concatenated into a text output using samtools view. BAM files were used to call variants using mpileup and normalised with norm commands in bcftools, the vcf was indexed and a consensus fasta was produced using bcftools consensus. rRNA position was predicted using barrnap, sorted by p value and extracted using the script '16S_sequence_Barrnap.sh' (https://github.com/raymondkiu/16S_extraction_Barrnap). The extracted 16S sequence was then BLASTed against NCBI non-redundant nucleotide database and the top hits downloaded. To extract sequences isolated from Anophelines, NCBI BioSample was searched for *Anopheles* and Microbes selected. All sequences annotated from the same

bacterial species were downloaded and used in the alignment. The reads were then aligned using MUSCLE and phylogenies produced using Maximum Likelihood with default parameters and 1000 bootstraps in MEGA X [79]. Metadata for the microbiome is detailed in S6 Table.

## Microbiome analysis

Primers were taken from previously published literature or designed using NCBI Primer-BLAST (S7 Table). To confirm the specificity of the primers, PCR was run using DreamTaq (Thermo) with the following cycle: 95˚C 2 min, 95˚C 30s, 60˚C 30s, 72˚C 20s, 72˚C 7min for 35 cycles. Following positive PCR, bands were extracted using QiaQuick Gel Extraction kit (Qiagen) following manufacturer's instructions and sent for Sanger sequencing using forward and reverse primers. For the extractions, mosquitoes were surface sterilised by submersion in 100% ethanol and allowed to dry before being mechanically disrupted in STE buffer, boiled at 95˚C for 10 minutes, centrifuged and supernatant removed.

## Supporting information

**S1 Fig. PCA of RNAseq datasets.** PCA performed on variance stabilising transformation on count data using DEseq2.
(TIF)

**S2 Fig. TaqMan assay of detoxification families.** Relative mRNA expression levels between the original (green) and susceptible (blue) and re-selected (orange) and susceptible (blue) for 8 genes previously linked with resistance. Significance was calculated by an ANOVA followed by Dunnett's multiple testing. Adjusted p values are shown with significance as follows: ** $p < 0.01$ and *** $p < 0.001$.
(TIF)

**S3 Fig. Mitochondrial read depth.** Read depth (y) along the full mitochondrial genome (x) for the re-selected (red) and susceptible (black).
(TIF)

**S4 Fig. Bray-Curtis dissimilarity.** Graph showing the similarities of the different samples in terms of microbe abundance.
(TIF)

**S5 Fig. Relative abundance of bacteria.** Log10 abundance of each bacteria meeting the cut-off criteria, were compared using a Mann Whitney test. In each case * $p < 0.05$, ** $p < 0.01$, *** $p < 0.001$, **** $p < 0.0001$.
(TIF)

**S6 Fig. 16S phylogenies of the most abundant bacteria.** Phylogenies constructed using 16S from the consensus genome for (a) *Elizabethkingia*; (b) *Asaia* and (c) *Serratia*. Each sequence with a green dot is confirmed to have been isolated from an Anopheline mosquito. Sequence extracted from the bacteria in this study are highlighted in red. Sequence names are taken directly from NCBI. Phylogenies created with MegaX using CLUSTAL alignment followed by a Neighbour Joining Tree with 1000 bootstraps. Figures on branches represent bootstrap values.
(PDF)

**S1 Table. RNAseq results significantly differential in both Banfora-R an Banfora-O comparisons and associated enrichment analysis.**
(XLSX)

**S2 Table. RNAseq data for the genes found to be duplicated within this population.** Gene ID, adjusted p and fold change for the original population compared to the susceptible population and adjusted p and fold change for the reselected compared to the susceptible population. Duplication ID taken from Lucas et al.
(XLSX)

**S3 Table. Description of GWAS-related SNPs and RNAseq block.**
(XLSX)

**S4 Table. Outlier genetic differences among colonies.**
(XLSX)

**S5 Table. BLAST results for unaligned predicted ORFs.**
(XLSX)

**S6 Table. Metadata for microbiome databases.**
(XLSX)

**S7 Table. Primers for bacterial PCR.** F indicates the forward primer and R the reverse. Source shows the literature source the primer was taken from or the NCBI genome reference used to design primers.
(XLSX)

**S1 Appendix. Description of other significant genomic regions.**
(DOCX)

## Acknowledgments

The authors would like to thank Manuela Bernardi for graphical elements used to create Fig 7, Professor Joanne Knight, Lancaster Medical School, for input in the GWAS analysis and Ms Antonia Böhmert for phylogeny construction.

## Author Contributions

**Conceptualization:** Victoria A. Ingham, Hilary Ranson.

**Data curation:** Victoria A. Ingham, Jacob A. Tennessen.

**Formal analysis:** Victoria A. Ingham, Jacob A. Tennessen, Eric R. Lucas.

**Funding acquisition:** Victoria A. Ingham, Grant L. Hughes, Eva Heinz, Daniel E. Neafsey.

**Investigation:** Victoria A. Ingham, Jacob A. Tennessen, Eric R. Lucas, Sara Elg, Henrietta Carrington Yates, Jessica Carson, Hilary Ranson.

**Methodology:** Victoria A. Ingham, Jacob A. Tennessen, Grant L. Hughes, Eva Heinz, Daniel E. Neafsey, Hilary Ranson.

**Project administration:** Victoria A. Ingham.

**Resources:** Victoria A. Ingham, Hilary Ranson.

**Software:** Victoria A. Ingham.

**Supervision:** Victoria A. Ingham, Wamdaogo Moussa Guelbeogo, N'Fale Sagnon, Hilary Ranson.

**Validation:** Victoria A. Ingham.

**Visualization:** Victoria A. Ingham.

**Writing – original draft:** Victoria A. Ingham, Hilary Ranson.

**Writing – review & editing:** Victoria A. Ingham, Hilary Ranson.

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
