## [Decision Letter · Decision Letter 0]

18 Oct 2021

Dear Dr Ingham,

Thank you very much for submitting your Research Article entitled 'Integration of whole genome sequencing and transcriptomics reveals a complex picture of insecticide resistance in the major malaria vector Anopheles coluzzii' to PLOS Genetics.

The manuscript was fully evaluated at the editorial level and by independent peer reviewers. The reviewers appreciated the attention to an important topic but identified some concerns that we ask you address in a revised manuscript

We therefore ask you to modify the manuscript according to the review recommendations. Your revisions should address the specific points made by each reviewer.

[LINK]

Yours sincerely,

Giorgio Sirugo

Associate Editor

PLOS Genetics

Gregory P. Copenhaver

Editor-in-Chief

PLOS Genetics

Reviewer's Responses to Questions

**Comments to the Authors:**

Reviewer #1: The manuscript provides an interesting and in-depth examination of a range of factors that are usually considered separately when examining insecticide resistance. The authors took a sensible research route and had a unique opportunity to examine resistance dynamics which may not be available for examination many times again.

The background provides enough clear detail and context to the study. The methodology is sound, with enough detail to allow replication. The statistical analysis is also described suitably. The mining of whole genome sequencing data is useful, particularly for the microbiome data as this circumvents the shortcomings of 16s sequencing.

My critiques are therefore minor.

1. It is interesting that your resistant strain is physically smaller than the susceptible. This is rather unusual, as it is usually opposite way around and it is thought that vigour tolerance could play a part in resistance phenotype. This should be mentioned at least briefly in the discussion.

2. Although I think I had understood what you were saying in the text before it, taken on its own, what you are trying to convey in figure 7. The legend does not stand alone and needs to be stated a bit more clearly.

3. Supplementary figure 4: this would benefit from the statistical indicators being present on the Bray Curtis dissimilarity plot.

4. Although the microbiome data is not the sole focus of the study, I think that it does add to the field of the mosquito microbiome data. I would suggest that one additional supplement would be the inclusion of the associated metadata as listed by the mosquito microbiome project:

https://mosquito-microbiome.org/files/Metadata_checklist.pdf

5. The one section that I think can be improved is the discussion. While there is nothing wrong with the statements or conclusions, it is a little descriptive, and I think you are missing an opportunity to reflect on a truly wonderful piece of research. What I would have been interested in as a reader (and I would have thought this would have been touched on having just read the abstract) is the significance. I am sure I am not the only mosquito researchers have suspected some of the findings that you have so elegantly displayed. You have shown that transcription changes did not have as dramatic changes in the resistance phenotype as would have been thought. What is the significance of this? Talk about genomic surveillance is everywhere right now. What does your findings mean for genomic surveillance of resistance? Does it have any bearing or does it only apply to your set up. This is just one example, but there is a lot more opportunity for you to get into the meat of your findings, because you have a lot of results to discuss.

Reviewer #2: In this manuscript, the authors used RNAseq and whole genome sequencing to identify changes within the mosquito’s genome, transcriptome and microbiome due to different resistance phenotypes. In the study, a susceptible and two resistant populations, having the same genetic background were used, which, as author claim allow better comparing such variables.

This article is well-written showing few spelling mistakes that can easily be corrected.

In the methods chapter, authors must clarify, at the RNA analysis section, which/how many replicates were in fact used, for each sample. Despite being noticiable in other points of the article it’s necessary to clarify it also here, in the “methods”. Again, at the methods chapter, please identify completely all the reagents and kits used and the same for relevant equipment. The methods and software must be properly referenced.

Regarding the results’s chapter please indicate what does it means “PCA” at the first time it is mentioned.

In order to study the mechanisms of resistance in An. coluzzi, 2 insecticide-resistant populations were used; although, throughout the text there are some references, it is necessary to emphasize the importance of using these two populations. In other words, it must be demonstrated that the inclusion of these populations in the trials was, without a doubt, important and not, on the contrary, unnecessary. So, please mention the relevant differences you found in these two populations showing, as claimed, that their inclusion “allowed a unique comparison of resistant and susceptible mosquitoes with the same genetic background”.

This article reached an excellent amount of results which would expect a more in-depth discussion. Please include more previous works in order to enrich this chapter.

As this paper shows a lot work regarding the changes in resistance restoration, it would benefit the article if this subject was mentioned also in the title.

**Have all data underlying the figures and results presented in the manuscript been provided?**

Reviewer #1: Yes

Reviewer #2: Yes

PLOS authors have the option to publish the peer review history of their article (what does this mean?). If published, this will include your full peer review and any attached files.

Reviewer #1: No

Reviewer #2: **Yes: **Ana Gonçalves Domingos

---

## [Decision Letter · Decision Letter 1]

27 Nov 2021

Dear Dr Ingham,

We are pleased to inform you that your manuscript entitled "Integration of whole genome sequencing and transcriptomics reveals a complex picture of the reestablishment of insecticide resistance in the major malaria vector Anopheles coluzzii" has been editorially accepted for publication in PLOS Genetics. Congratulations!

Yours sincerely,

Giorgio Sirugo

Associate Editor

PLOS Genetics

Gregory P. Copenhaver

Editor-in-Chief

PLOS Genetics

Comments from the reviewers (if applicable):

Reviewer's Responses to Questions

**Comments to the Authors:**

Reviewer #1: The authors have attended well to the requested revisions, and the documents should be published. Please just note the additional request for a reference to be added to the discretion of the editor.

Reviewer #2: Thie manuscript version now submitted responds to the questions and suggestions requested. The manuscript is therefore ready for publication.

**Have all data underlying the figures and results presented in the manuscript been provided?**

Reviewer #1: Yes

Reviewer #2: Yes

PLOS authors have the option to publish the peer review history of their article (what does this mean?). If published, this will include your full peer review and any attached files.

Reviewer #1: No

Reviewer #2: **Yes: **Ana Gonçalves Domingos

**Data Deposition**

http://datadryad.org/submit?journalID=pgenetics&manu=PGENETICS-D-21-01204R1

**Press Queries**

---

## [Editor Report · Acceptance letter]

15 Dec 2021

PGENETICS-D-21-01204R1 

Integration of whole genome sequencing and transcriptomics reveals a complex picture of the reestablishment of insecticide resistance in the major malaria vector Anopheles coluzzii 

Dear Dr Ingham, 

We are pleased to inform you that your manuscript entitled "Integration of whole genome sequencing and transcriptomics reveals a complex picture of the reestablishment of insecticide resistance in the major malaria vector Anopheles coluzzii" has been formally accepted for publication in PLOS Genetics! Your manuscript is now with our production department and you will be notified of the publication date in due course.

With kind regards,

Zsofia Freund

PLOS Genetics

On behalf of:
